# Quantitative Analysis of Super Resolution in Electromagnetic Inverse Scattering for Microwave Medical Sensing and Imaging

**DOI:** 10.3390/s23177404

**Published:** 2023-08-25

**Authors:** Yahui Ding, Zheng Gong, Yifan Chen, Jun Hu, Yongpin Chen

**Affiliations:** 1School of Electronic Science and Engineering, University of Electronic Science and Technology of China, Chengdu 611731, China; dingyahui@std.uestc.edu.cn (Y.D.); hujun@uestc.edu.cn (J.H.); ypchen@uestc.edu.cn (Y.C.); 2Yangtze Delta Region Institute (Quzhou), University of Electronic Science and Technology of China, Quzhou 324003, China; gong.z@outlook.com; 3School of Life Sciences and Technology, University of Electronic Science and Technology of China, Chengdu 611731, China

**Keywords:** electromagnetic inverse problems, resolution, super-resolution, point spread function, Rayleigh criterion, Sparrow criterion

## Abstract

Microwave medical sensing and imaging (MMSI) has been a research hotspot in the past years. Imaging algorithms based on electromagnetic inverse scattering (EIS) play a key role in MMSI due to the super-resolution phenomenon. EIS problems generally employ far-field scattered data to reconstruct images. However, the far-field data do not include information outside the Ewald’s sphere, so theoretically it is impossible to achieve super resolution. The reason for super resolution has not been clarified. The majority of the current research focuses on how nonlinearity affects the super-resolution phenomena in EIS. However, the mechanism of super-resolution in the absence of nonlinearity is routinely ignored. In this research, we address a prevalent yet overlooked problem where the image resolution due to scatterers of extended structures is incorrectly analyzed using the model of point scatterers. Specifically, the classical resolution of EIS is defined by the Rayleigh criterion which is only suitable for point-like scatterers. However, the super-resolution in EIS is often observed for general scatterers like cylinders, squares or Austria shapes. Subsequently, we provide theoretical results for the Born approximation framework in EIS, and employ the Sparrow criteria to quantify the resolution for symmetric objects of extended structures. Furthermore, the modified Sparrow criterion is proposed to calculate the resolution of asymmetric scatterers. Numerical examples show that the proposed approach can better explain the super-resolution phenomenon in EIS.

## 1. Introduction

Microwave medical sensing and imaging (MMSI) uses non-ionizing electromagnetic waves at microwave frequencies for the diagnosis of functional and pathological tissue conditions. The applications of MMSI include breast cancer screening, brain stroke diagnosis, and bone fracture imaging [1,2,3,4,5,6,7,8]. Compared with traditional medical imaging methods such as Computed Tomography (CT) and Magnetic Resonance Imaging (MRI), MMSI systems are more portable, lower cost and reduce health risks. For imaging techniques, both quantitative and qualitative imaging algorithms have been developed. The former are mainly based on electromagnetic inverse scattering (EIS) whereas the latter utilize radar-based approaches.

The objective of EIS is to reconstruct the dielectric profile of scatterers, which can be used to sense anomalies with distinctive dielectric properties. In this paper, we mainly focus on the case of using far-field scattered data for dielectric reconstruction, as commonly studied in EIS. The resolution is a central metric in EIS for determining the sensing system’s capability. For instance, the resolution in [9] is about 1/12 wavelength, which achieves super-resolution performance because the result significantly exceeds the diffraction limit (1/2 wavelength). However, the reason behind super-resolution in EIS has not been completely understood. According to the analytical results using the spectrum theory [10], the imaging resolution based on Born approximation (BA) can only reach about 0.3 wavelength. Subsequently, researchers proposed the Born approximation iteration method (BIM) to improve the results of BA, and argued that this iterative scheme accounts for multiple scattering of electromagnetic waves. Hence, they interpreted super-resolution as a result of multiple scattering of the incident wave, which transforms the evanescent wave to the transmission wave that radiates to the far field. The evanescent wave contains information outside the Ewald sphere resulting in super-resolution. However, this explanation is subject to debate. For instance, ref. [10] points out that super-resolution can be achieved without multiple scattering. Moreover, ref. [11] suggests that even if the far field contains the information outside the Ewald sphere, it cannot naturally lead to the super-resolution phenomenon. This is because even though multiple scattering transforms the evanescent wave into the transmission wave, the latter is destroyed and the spectrum of the former should be obtained which is hard to achieve due to complex multiple scattering. Also, a recent paper [12] argued that such multiple scattering would lead to 0/0 manifolds from a mathematical point of view, which would make it meaningless to yield super-resolution. The existing literature mainly focuses on whether multiple scattering leads to super-resolution. Nevertheless, few works are dedicated to super-resolution without multiple scattering. In [13], super-resolution was interpreted as the result of analytical continuation, which however is not feasible in practice due to its serious instability and high sensitivity to noise as mentioned in [14]. Consequently, for weak scattering problems, we still need new tools to explain the super-resolution characteristics.

In this paper, we extend the definition of resolution to offer new insight into super-resolution. The diffraction limit in optics is calculated by the Rayleigh criterion, which is defined such that the centre of the diffraction pattern of one image is directly over the first minimum of the diffraction pattern of the other. Using this definition, we can obtain a resolution limit of 0.383 wavelength. For point scatterers, the result is correct. When the distance between two points is less than 0.383 wavelength, it is impossible to distinguish them from the reconstructed image. However, the result becomes inaccurate if general scatterers such as two cylinders are employed. We can find that the resolution would be reduced with an increased scatterer radius. Hence, the Rayleigh criterion cannot be applied to general scatterers directly. To deal with this problem, there are two problems to be solved. The first problem is how to describe the spread function of general scatterers; the second one is how to determine whether the scatterers on the reconstructed image could be resolved. For the first problem, we review the BA framework and derive the corresponding point spread function (PSF). Subsequently, the generalized spread function (GSF) for a general scatterer can be regarded as the sum of PSFs if the scatterers are weak. For the second problem, we consider the Sparrow criterion traditionally used in astronomy. It is defined as the separation distance between two targets when the joint function has no dip in intensity at the midpoint. By combining the GSF and the Sparrow criterion, we propose a novel resolution computation method as well as a new interpretation of super-resolution in EIS under weak scattering. The reason behind super-resolution by BA is not that we get the information ouside the Ewald’s sphere.On the contrary, we found that the definition of super resolution is the key factor. With the new definition, the size of the scatterers have a great impact on the resolution. When the scatterers are points, super-resolution does not occur. But when the scatterers become larger, the resolution calculated by the Sparrow criterion can reach super-resolution. Moreover, most real-life scatterers are asymmetrical. In order to deal with asymmetry, we further propose a modified Sparrow criterion. In this case, the relative permittivity of scatterers have the same effect on the resolution. Numerical experiments verify the validity of the proposed methods. In short, we believe that the super-resolution is related to the size and the relative permittivity of the scatterers.

This paper proposes a new perspective for interpreting the super-resolution phenomenon in EIS by employing the Sparrow criterion, instead of the conventional Rayleigh criterion. In order to achieve this, the explicit expression of the spread function for general scatterers in EIS needs to be obtained. Thus, the BA method is first investigated. Subsequently, the Sparrow criterion is used to calculate the resolution of symmetrical scatterers and is modified to accommodate asymmetrical scatterers. Following this method, the size of the scatterer would also affect the resolution. As the cylindrical scatterer gradually increases in size, the image resolution is improved. The proposed analysis can be readily extended to iterative methods such as BIM and DBIM.

The paper is organized as follows. Section 2 briefly reviews the general framework of EIS under BA. Section 3 gives the spread function of BA. Section 4 analyzes the resolution of EIS by applying the Sparrow criterion. Numerical examples are then used to validate the analysis. Finally, concluding remarks are provided in Section 5.

## 2. General Framework of EIS

We consider a two-dimensional (2D) scenario with transverse magnetic wave illumination. As shown in Figure 1, the unknown scatterers are located in the domain of interest (**D**). Without loss of generality, the antennas are supposed to be distributed on the circumference of a circle (**S**). The background is homogeneous. The equations describing the relationship between the scatterers and the field can be written as,
(1)$$J=\chi (E^{i}+G^{D}J),$$
and
(2)$$E^{s}=G^{S}J,$$
where J is the contrast current; χ is the contrast function defined as $\chi \left(r\right)=\left[\epsilon \left(r\right)-\epsilon_{0}\right]/\epsilon_{0}$ with ϵ(r) and $\epsilon_{0}$ being the permittivities of scatterer r and background, respectively; $E^{i}$ is the incident field; $G^{D}$ and $G^{S}$ are the matrix forms of the Green functions in domain **D** and surface **S**, respectively; and $E^{s}$ is the scattering field. It is worth mentioning that our focus is on the resolution generated in far field, thus the receiving antenna is located in the far field area.

In solving the EIS problems, the incident field $E^{i}$, the scattering field $E^{s}$, and the Green function matrixes $G^{D}$ and $G^{S}$ are known. The contrast function χ and the current J are unknown. For the BA method, the first step is to obtain the current J using (Equation 2). This yields an underdetermined equation with the following solution,
(3)$$J^{sol}={\left(G^{S}\right)}^{*}{\left[G^{S}{\left(G^{S}\right)}^{*}\right]}^{-1}E^{s},$$
where ${\left(\cdot \right)}^{*}$ denotes the Hermitian conjugate and $J^{sol}$ is the solution of (Equation 2) that minimizes the norm $\left|\left|J\right|\right|$, where $\left|\left|J\right|\right|$ is the 2-norm of J.

The second step is to calculate the contrast function χ using (Equation 1). In consideration of BA, the scattering field is very weak compared with the incident field. Therefore, it can be approximated as $J=\chi E^{i}$. This equation is overdetermined without an exact solution. A common method is to obtain the least squares solution,
(4)$$\chi^{sol}=\frac{\sum_{n=1}^{N^{i}}J_{n}^{sol}\cdot {\left(E_{n}^{i}\right)}^{*}}{\sum_{n=1}^{N^{i}}E_{n}^{i}\cdot {\left(E_{n}^{i}\right)}^{*}},$$
where the subscript *n* denotes the nth incident wave, $N^{i}$ is the total number of incident waves, and $\chi^{sol}$ is the solution of contrast function under BA.

In summary, BA can separate the inverse process into two steps. The first is to use the minimum norm method to obtain the current J, and the second is to use the least squares method to obtain the contrast function χ. As BA is only suitable for weak scatterers, researchers have developed the Born iterative method (BIM) to deal with strong scatterers. In each iteration of BIM, the first step is the same as BA, and the second step is to calculate the current by using the total field derived from χ. A further improved method is to calculate the current by using the Green function of an inhomogeneous medium, which is called the Distorted Born iterative method (DBIM) [15]. Other types of methods solve the current and contrast functions by using optimization algorithms [16,17,18].

## 3. General Spread Function

A key quantity in the examination of resolution is the spread function. In this section, we first re-derive the classical point spread function (PSF) from the perspective of BA, which allows us to extend it to the general spread function (GSF) for scatterers of extended structures. This analysis paves the way for the discussion of super-resolution based on various forms of Sparrow criteria in the following section.

Consider the scenario of point scatterers. The PSF can be derived from the process of BA. First, we can compute the current using (Equation 3). The current and the scattering field are linked by the Green’s function, which is the zeroth-order Hankel function of the first kind in 2D free space,
(5)$$E^{s}=k_{0}^{2}\int_{D}g\left(r,r^{\prime }\right)J\left(r^{\prime }\right)dr^{\prime },$$
where
(6)$$g\left(r,r^{\prime }\right)=-\frac{j}{4}H_{0}^{\left(1\right)}\left(k_{0}\left|r-r^{\prime }\right|\right),$$
with $H_{0}^{\left(1\right)}\left(\cdot \right)$ being the zeroth-order Hankel function of the first kind. $k_{0}$ is the wave number of incident wave. Because antennas are located in the far field, we can use the following approximation of Hankel function,
(7)$$\lim_{\left|r-r^{\prime }\right|\to \infty }H_{0}^{\left(1\right)}\left(k_{0}\left|r-r^{\prime }\right|\right)=\sqrt{\frac{2}{j\pi k_{0}\left|r-r^{\prime }\right|}}e^{jk_{0}\left|r-r^{\prime }\right|}.$$
Substitute (Equation 6) and (Equation 7) into (Equation 5) results in
(8)$$E^{s}=k_{0}^{2}\int_{-\infty }^{\infty }-\frac{1}{4}\sqrt{\frac{2j}{\pi k_{0}\left|r-r^{\prime }\right|}}e^{jk_{0}\left|r-r^{\prime }\right|}J\left(r^{\prime }\right)dr^{\prime }.$$
Suppose that antennas are located in the far field. In this case, r is much larger than $r^{\prime }$. The term $\left|r-r^{\prime }\right|$ in the denominator in (Equation 8) can be approximated by |r|. Furthermore, the term $\left|r-r^{\prime }\right|$ in the exponent can be approximated by $\left|r\right|-\left|r^{\prime }\right|\cos\theta$, where θ is the angle between r and $r^{\prime }$. Applying $\cos\theta =\frac{rr^{\prime }}{\left|r\right|\left|r^{\prime }\right|}$, it comes to $\left|r\right|-r^{\prime }\cdot e_{r}$, where $e_{r}$ is the unit vector in the r direction. Subsequently, we have
(9)$$E^{s}=-\frac{k_{0}^{2}}{4}\sqrt{\frac{2j}{\pi k_{0}\left|r\right|}}e^{jk_{0}|r|}\int_{-\infty }^{\infty }e^{-jk_{0}r^{\prime }\cdot e_{r}}J\left(r^{\prime }\right)dr^{\prime }.$$
Using the Cartesian coordinate system in (Equation 9), it comes to
(10)$$\begin{array}{cc} E^{s}= & -\frac{k_{0}^{2}}{4}\sqrt{\frac{2j}{\pi k_{0}\left|r\right|}}e^{jk_{0}\left|r\right|}\\ \; & \int_{-\infty }^{\infty }\int_{-\infty }^{\infty }e^{-j2\pi \left(\frac{k_{x}}{2\pi }x+\frac{k_{y}}{2\pi }y\right)}J\left(x,y\right)dxdy, \end{array}$$
where $k_{x}^{2}+k_{y}^{2}=k_{0}^{2}$. We can find that the double integral in (Equation 10) indicates the 2D Fourier transform of the current over a ring of radius $\frac{k_{0}}{2\pi }$. Defining a constant term $C=-\frac{k_{0}^{2}}{4}\sqrt{\frac{2j}{\pi k_{0}\left|r\right|}}e^{jk_{0}\left|r\right|}$, the scattering field reduces to
(11)$$E^{s}=C\cdot F\left(J\right)\cdot \mathrm{ring}\left(\frac{k_{0}}{2\pi }\right),$$
where F(·) denotes the 2D Fourier transform, and ring$\left(\frac{k_{0}}{2\pi }\right)$ is the 2D Dirac delta function with a ring shape of radius $\frac{k_{0}}{2\pi }$.

Next, we use the singular value decomposition (SVD) to decompose the current into a radiation component and a non-radiation component [18]. The SVD of $G^{S}$ is expressed as
(12)$$G^{S}=U\Sigma V^{*},$$
where U and V are composed of orthonormal left and right singular vectors, respectively. The term Σ is a diagonal matrix composed of singular values. In this form, the vector of the current can be written as a span of the right singular vectors V. And we can decompose the current into two complementary and orthogonal parts,
(13)$$J=J^{+}+J^{-},$$
where $J^{+}$ and $J^{-}$ are corresponding to the first and last columns of V, respectively. According to SVD theory, the radiation current $J^{+}$ produces the scattering field, whereas the non-radiation current $J^{-}$ has no contribution. Then the value of $J^{+}$ can be obtained by applying (Equation 2) and (Equation 12),
(14)$$J^{+}=V\frac{U^{*}\cdot E^{s}}{\Sigma }.$$
Applying the SVD theory, the pseudo inverse of $G^{S}$ in (Equation 3) can also be represented as
(15)$${\left(G^{S}\right)}^{*}{\left[G^{S}{\left(G^{S}\right)}^{*}\right]}^{-1}=V\Sigma^{-1}U^{*}.$$
So the current is given by
(16)$$J^{sol}=V\Sigma^{-1}U^{*}E^{s}=V\frac{{\left(U\right)}^{*}\cdot E^{s}}{\Sigma }=J^{+}.$$
Subsequently, the current in (Equation 11) can also decomposed into the form of (Equation 13), which leads to
(17)$$E^{s}=C\cdot F\left(J^{+}+J^{-}\right)\cdot \mathrm{ring}\left(\frac{k_{0}}{2\pi }\right).$$
On the basis of the definitions of radiation and non-radiation currents, the term $J^{+}$ is the value of F(J) on the ring,
(18)$$E^{s}=C\cdot F\left(J^{+}\right).$$
Hence, the inverse Fourier transform of the scattering field is $J^{+}$,
(19)$$J^{+}=\frac{1}{2\pi C}F^{-1}\left(E^{s}\right)$$
Substituting (Equation 11) and (Equation 19) into (Equation 16) yields
(20)$$\begin{array}{cc} J^{sol}=\frac{1}{2\pi C}F^{-1}\left(E^{s}\right) & =\frac{1}{2\pi C}F^{-1}\left[C\cdot F\left(J\right)\cdot \mathrm{ring}\left(\frac{k_{0}}{2\pi }\right)\right]\\ \; & =\frac{1}{2\pi }\left\{J*F\left[\mathrm{ring}\left(\frac{k_{0}}{2\pi }\right)\right]\right\}, \end{array}$$
where the symbol * denotes convolution. It is worth noting that the result would be zero if the ring had zero thickness. However, the quantization effect of the numerical computation in (Equation 3) results in a non-zero thickness of the ring implicitly. Hence, the result comes to
(21)$$J^{sol}=\frac{1}{2\pi }\left\{J*F\left[\mathrm{ring}\left(\frac{k_{0}}{2\pi },\Delta R\right)\right]\right\},$$
where ΔR is the effective thickness of the ring due to the quantization effect. Following the derivation of the 2D Fourier transform of a ring in Appendix A, the result is
(22)$$F\left[\mathrm{ring}\left(\frac{k_{0}}{2\pi },\Delta R\right)\right]=k_{0}\Delta RJ_{0}\left(k_{0}\rho \right),$$
where ρ represents the distance from the origin in the reconstructed image. Substituting (Equation 22) into (Equation 20) yields
(23)$$\begin{array}{cc} J^{sol} & =\frac{1}{2\pi }\left[J*k_{0}\Delta RJ_{0}\left(k_{0}\rho \right)\right]\\ \; & \propto J*J_{0}\left(k_{0}\rho \right). \end{array}$$

To reconstruct the relative permittivity profile, the least squares method in (Equation 4) is used. Applying the plane wave equation in (Equation 4), we have
(24)$$\chi^{sol}=\frac{1}{N^{i}}J^{sol}\sum_{n=1}^{N^{i}}e^{j(k_{0}\rho_{\theta_{n}})},$$
where $\theta_{n}$ is the incident angle of the nth plane wave. For a sufficiently large $N^{i}$, the summation can be approximated by an integral
(25)$$\chi^{sol}=\frac{1}{N^{i}}J^{sol}\int_{0}^{2\pi }e^{jk_{0}\rho_{\theta }}d\theta .$$
For simplicity, we focus on the value along the *x*-axis with y=0,
(26)$$\chi^{sol}\left(x\right)=\frac{1}{N^{i}}J^{sol}\int_{0}^{2\pi }e^{jk_{0}x\cos\theta }d\theta .$$
It can be found that the integral in (Equation 26) has the same expression as the zeroth-order Bessel function,
(27)$$J_{0}\left(k_{0}x\right)=\frac{1}{2\pi }\int_{0}^{2\pi }e^{jk_{0}x\cos\theta }d\theta .$$
Substitute (Equation 27) into (Equation 26) results in
(28)$$\chi^{sol}\left(x\right)=\frac{1}{2\pi N^{i}}J^{sol}J_{0}\left(k_{0}x\right).$$
We can then derive the PSF from (Equation 23) and (Equation 28),
(29)$$\chi^{sol}\left(x\right)\propto {\left[J_{0}\left(k_{0}x\right)\right]}^{2}.$$

To verify the above result, the reconstructed image of a point scatterer at 300 MHz is shown in Figure 2a. Furthermore, Figure 2b compares the normalized amplitude of the reconstructed image along the *x*-axis and the PSF in (Equation 29), which shows excellent agreement between these two functions.

For a general scatterer, the reconstructed image could be regarded as the superposition of PSFs for all point scatterers that comprise the general scatterer if the mutual coupling of these individual point scatterers could be ignored. Hence, we can extend the concept of PSF to a general scatterer, which results in the following approximation
(30)$$\chi^{sol}\left(r\right)\propto \chi \left(r\right)*{\left[J_{0}\left(k_{0}r\right)\right]}^{2}.$$
Equation (Equation 30) defines the general scatterer spread function (GSF). Although it is an approximate solution under BA, it can be regarded as an upper limit in any chosen frequency.

## 4. Resolution Analyses Using Various Forms of Sparrow Criteria

### 4.1. Standard Sparrow Criterion

#### 4.1.1. Point Scatterers

We first review the classical Rayleigh criterion. It states that two images become resolvable when the centre of the diffraction pattern of one image is directly over the first minimum of the diffraction pattern of the other. The resolution defined by the Rayleigh criterion can be expressed as
(31)$$\sigma =\min\left|\arg_{x}\left\{\chi^{sol}\left(x\right)=0\right\}\right|.$$
Substituting (Equation 29) into (Equation 31) results in σ=0.383λ.

Then, the standard Sparrow criterion is applied, which is defined as the separation distance between two targets when the joint function has no dip in intensity at the midpoint. The resolution defined by this criterion is
(32)$$\sigma =\min\left|\arg_{x}\left\{\frac{d^{2}\chi^{sol}\left(x\right)}{dx^{2}}=0\right\}\right|,$$
where $\chi^{sol}\left(x\right)$ is the overall spread function along the *x*-axis with y=0.

The resolutions obtained using the criteria in (Equation 31) and (Equation 32) are 0.383λ and 0.344λ, respectively. In Figure 3, we plot the reconstructed images and the sectional view of two points with separation distances of 0.30 m (Figure 3a–c), 0.34 m (Figure 3d–f) and 0.38 m (Figure 3g–i). As can be seen from the figure, the two points in Figure 3a) are not resolved as only one peak is observed in Figure 3c). On the other hand, in Figure 3i, we can clearly observe a slight trough between two peaks, which indicates that the two points in Figure 3g have been resolved.

#### 4.1.2. Cylinder Scatterers

In general, the definition of resolution is determined by the separation from the scatterer’s edges. Hence, the definition of resolution comes to
(33)$$\sigma =\min\left|\arg_{x}\left\{\frac{d^{2}\chi^{sol}\left(x\right)}{dx^{2}}=0\right\}\right|-2a,$$
where *a* is the distance from the edge to the center of each scatterer.

For illustration purposes, we consider two cylinder scatterers. In this case, $\chi^{sol}\left(x\right)$ can be written as
(34)$$\begin{array}{cc} \chi^{sol}\left(x\right)= & \int_{x-d/2-r}^{x-d/2+r}{\left[J_{0}\left(k_{0}\rho \right)\right]}^{2}d\rho \\ \; & +\int_{x+d/2-r}^{x+d/2+r}{\left[J_{0}\left(k_{0}\rho \right)\right]}^{2}d\rho , \end{array}$$
where *r* is the radius of each circle and *d* is the distance between the centers of the circles. According to the definition of Sparrow criterion, we need to calculate the first null point of the second derivative of the joint function when x=0. This is given by
(35)$$\frac{d^{2}\chi^{sol}\left(x\right)}{dx^{2}}=\begin{array}{cc} \; & 2k_{0}J_{0}\left[k_{0}\left(x+d/2-r\right)\right]J_{1}\left[k_{0}\left(x+d/2-r\right)\right]\\ \; & -2k_{0}J_{0}\left[k_{0}\left(x+d/2+r\right)\right]J_{1}\left[k_{0}\left(x+d/2+r\right)\right]\\ \; & +2k_{0}J_{0}\left[k_{0}\left(x-d/2-r\right)\right]J_{1}\left[k_{0}\left(x-d/2-r\right)\right]\\ \; & -2k_{0}J_{0}\left[k_{0}\left(x-d/2+r\right)\right]J_{1}\left[k_{0}\left(x-d/2+r\right)\right], \end{array}$$
and the resolution is obtained as
(36)$$\sigma =\min\left|\arg_{x}\left\{\frac{d^{2}\chi^{sol}\left(x\right)}{dx^{2}}=0\right\}\right|-2r.$$

For example, if we set the frequency as 300 MHz and the radius as 0.1 m, the resolution is obtained as 0.152 m. Figure 4a,b are the actual and reconstructed images for a separation distance of 0.12 m. Figure 4d,e are the images for a separation distance of 0.16 m. Figure 4g,h are the images for a separation distance of 0.2 m. Figure 4c,f,i are the sectional views of the reconstructed images. We can find that the two scatterers can be resolved at 0.2 m. It is hard to say the two cylinders are resolved at 0.16 m. This result is because the distance is too close to the calculation result. But we believe the calculation results indicate the demarcation between whether it can be distinguished. Furthermore, as $\frac{\sigma }{\lambda }=0.152$, the resolution with respect to wavelength is around 0.152λ. Hence, the Sparrow criterion could explain the phenomenon of super resolution. In addition, if we want to achieve the minimum level of super-resolution (0.25λ), we can find that the radius of cylinder should be 0.048λ.

### 4.2. Modified Sparrow Criterion

To deal with different asymmetric scatterers, the Sparrow criterion should be modified. The Gaussian point between two scatterers is used to deal with this situation. First, the Gaussian point is calculated as
(37)$$g=\min\left|\arg_{x}\left\{\frac{d\chi^{sol}\left(x\right)}{dx}=0\right\}\right|,$$
And the resolution comes to
(38)$$\sigma =\min\left|\arg_{x}\left\{\frac{d^{2}\chi^{sol}\left(x\right)}{dx^{2}}=g\right\}\right|-a_{1}-a_{2}.$$
where $a_{1}$ and $a_{2}$ are the edge-to-center distances of the two scatterers, respectively.

We first consider two point scatterers with different permittivity. We set the relative permittivity values of two points as 1.1 and 1.2, respectively, and the frequency as 300 MHz. In Figure 5, we plot the reconstructed images and the sectional view of two points with separation distances of 0.34 m (Figure 5a–c), 0.38 m (Figure 5d–f) and 0.42 m (Figure 5g–i). The numerical results of resolution by (Equation 37) and (Equation 38) is 0.382 m. Together with Figure 3, we can find that the resolution is reduced if the permittivity values of the two points are different.

Next, we look into two cylindrical scatterers with different radii. The frequency is set as 300 MHz, and the radii of two cylinders are 0.1 m and 0.15 m. Figure 6a,b are the actual and reconstructed images when the distance of separation is 0.11 m. Figure 6d,e are the actual and reconstructed images when the distance is 0.15 m. Figure 6g,h are the actual and reconstructed images when the distance of separation is 0.19 m. Figure 6c,f,i are the sectional views of the reconstructed images. The numerical results of resolution is 0.147 m. The two cylinders can be resolved in 0.15 m and 0.19 m. Numerical results verify the accuracy of the generalized Sparrow criterion.

## 5. Conclusions

Resolution limit is an important topic in EIS. Nevertheless, the determination of resolution has been subject to debate. Most of the existing works on super-resolution focus on the role of multiple scattering, whereas super-resolution without involving multiple scattering is often ignored. The traditional PSF and Rayleigh criterion cannot explain super-resolution in the latter case. To address these issues, we proposed to use the GSF and Sparrow criterion to calculate the resolution. We found that two point scatterers can be resolved for a distance of 0.344λ. The resolution would be improved if two cylindrical scatterers were employed. The super-resolution would be observed if the cylinder radius was more than 0.048λ.

## Figures and Tables

**Figure 1 sensors-23-07404-f001:**
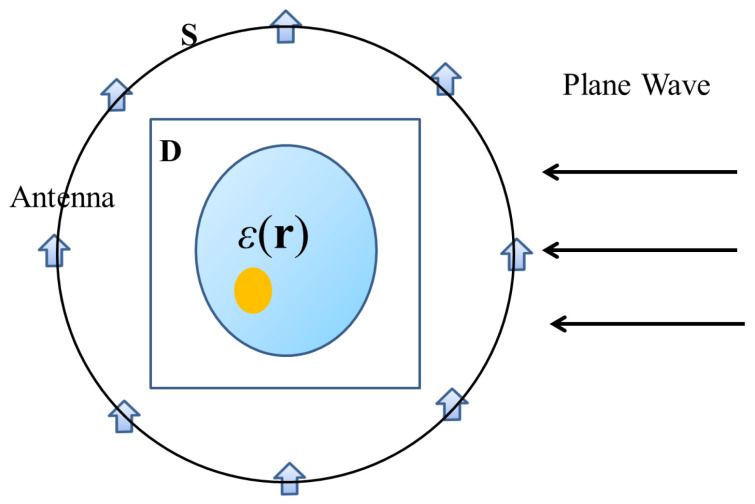
Schematic diagram of EIS.

**Figure 2 sensors-23-07404-f002:**
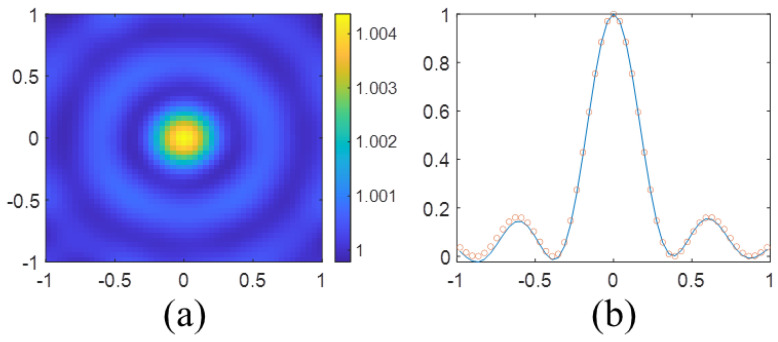
(**a**) Reconstructed image of a point scatterer in 300 MHz. (**b**) Comparison of the normalized amplitude of the reconstructed image along the *x*-axis and the PSF in (Equation 29).

**Figure 3 sensors-23-07404-f003:**
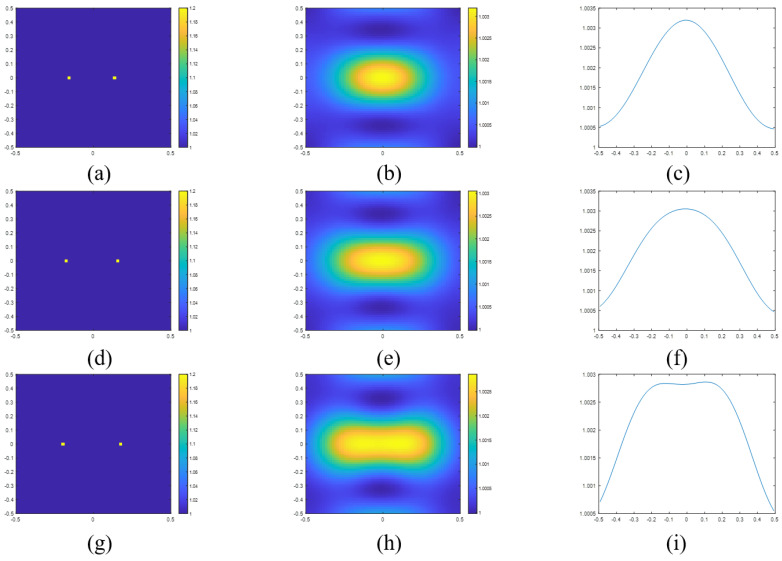
(**a**) Actual permittivity profile of two point scatterers with a separation distance of 0.30 m and the corresponding (**b**) reconstructed image and (**c**) the sectional view in the x-axle of reconstructed image. (**d**) Actual permittivity profile of two point scatterers with a separation distance of 0.34 m and the corresponding (**e**) reconstructed image and (**f**) the sectional view in the x-axle of reconstructed image. (**g**) Actual permittivity profile of two point scatterers with a separation distance of 0.38 m and the corresponding (**h**) reconstructed image and (**i**) the sectional view in the x-axle of reconstructed image.

**Figure 4 sensors-23-07404-f004:**
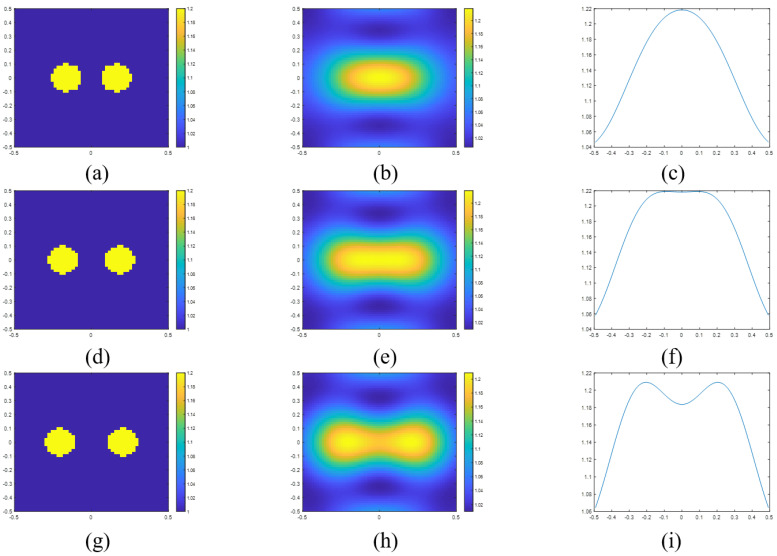
(**a**) Actual permittivity profile of two scatterers with a separation distance of 0.12 m and the corresponding (**b**) reconstructed image and (**c**) the sectional view in the x-axle of reconstructed image. (**d**) Actual permittivity profile of two scatterers with a separation distance of 0.16 m and the corresponding (**e**) reconstructed image and (**f**) the sectional view in the x-axle of reconstructed image. (**g**) Actual permittivity profile of two scatterers with a separation distance of 0.20 m and the corresponding (**h**) reconstructed image and (**i**) the sectional view in the x-axle of reconstructed image.

**Figure 5 sensors-23-07404-f005:**
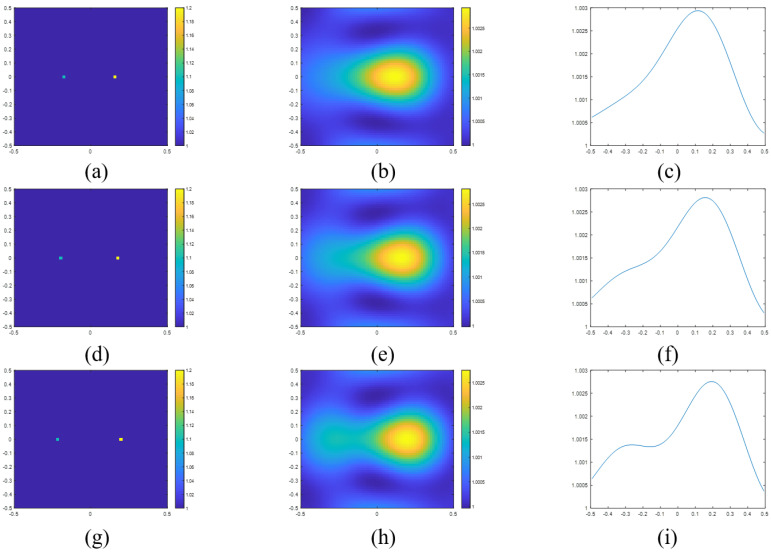
(**a**) Actual permittivity profile of two scatterers with a separation distance of 0.34 m and the corresponding (**b**) reconstructed image and (**c**) the sectional view in the x-axle of reconstructed image. (**d**) Actual permittivity profile of two scatterers with a separation distance of 0.38 m and the corresponding (**e**) reconstructed image and (**f**) the sectional view in the x-axle of reconstructed image (**g**) Actual permittivity profile of two scatterers with a separation distance of 0.42 m and the corresponding (**h**) reconstructed image and (**i**) the sectional view in the x-axle of reconstructed image.

**Figure 6 sensors-23-07404-f006:**
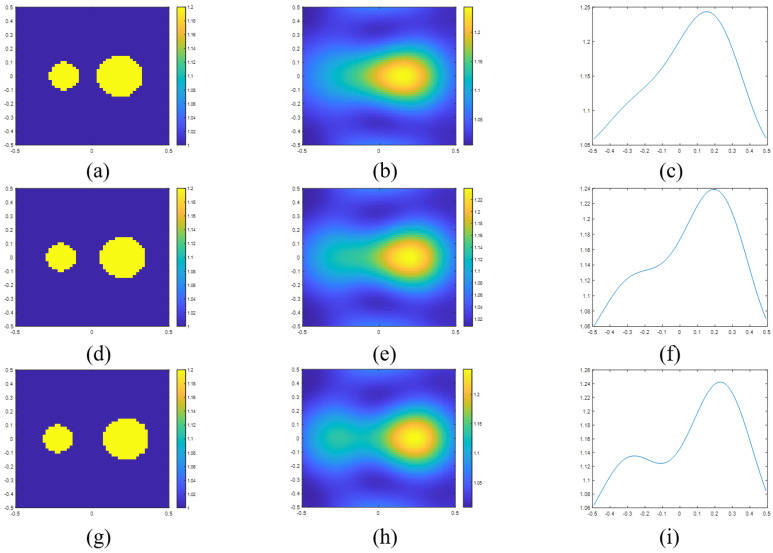
(**a**) Actual permittivity profile of two scatterers with a separation distance of 0.11 m and the corresponding (**b**) reconstructed image and (**c**) the sectional view in the x-axle of reconstructed image. (**d**) Actual permittivity profile of two scatterers with a separation distance of 0.15 m and the corresponding (**e**) reconstructed image and (**f**) the sectional view in the x-axle of reconstructed image. (**g**) Actual permittivity profile of two scatterers with a separation distance of 0.19 m and the corresponding (**h**) reconstructed image and (**i**) the sectional view in the x-axle of reconstructed image.

## Data Availability

Not applicable.

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
