# Peer review of "Quantitative Analysis of Super Resolution in Electromagnetic Inverse Scattering for Microwave Medical Sensing and Imaging"

_sensors, 2023, doi:10.3390/s23177404_

Round 1
Reviewer 1 Report
Dear Authors,
The work is of great interest and is appropriately presented. I have questions about some things I could have missed while reading the work. The proposed method to check the resolution is interesting.
- The approach is thought for the far field. This condition should be stated already in the abstract and introduction, not only from the beginning of section 2.
- What do the authors think would be the effect of doing this analysis in the near field?
- The authors consider plane waves as the excitation of the region of interest. Which would be the effect of using radiation points?
- The dielectric properties are slightly higher than the background. Which is the effect on the resolution of higher dielectric values? It does only affect if we have multiple targets with different sizes or dielectric values?
- The authors present figures 3.f. and 4.d as "resolved targets." In both cases, I see that the central part equals the maximum value of dielectric properties (long axis of the oval with no minimum or lowering). How can we say from this reconstruction that there are separate targets?
- The authors say in the introduction that they will explain the reason for having super-resolution, but it seems they are only showing some cases where it is observed, with some guesses. Could the authors comment on this?
Reviewer 2 Report
The authors discuss the resolution in the Born approximation in this manuscript. The discussion about resolution of inverse scattering is very important research topic. The Born approximation can accurately estimate the relative permittivity of distributed targets in the case of low contrast. However, the reconstruction results provided by authors based on the Born approximation are very different from the true values in Figs. 3, 4, and 5. Therefore, it is impossible to judge whether the authors' claims are correct or not.
Round 2
Reviewer 2 Report
There are two methods for imaging an object using microwaves. One is the radar method. It images the reflection coefficient distribution of the object. Another method is microwave tomography. The microwave tomography creates an image of electrical parameter distribution with respect to the relative permittivity and conductivity of the object. The reflection coefficient is second-order quantity of the electrical parameters (permittivity and conductivity). Therefore, microwave tomography has a high difficulty for creating images as compared with the radar.
A purpose of microwave tomography is to reconstruct the electrical parameter distribution of the object.
For microwave tomography, a method using the Born approximation was first proposed. However, it could only reconstruct low-contrast objects. Since the actual targets have the high contrast property (medical imaging, non-destructive examination, land mine detection, etc), iterative method and other methods have been proposed. Among them, a resolution was also one of the research subjects. For example, there is one paper about resolution of microwave toography (Colin Gilmore, Member, Puyan Mojabi, Amer Zakaria, Stephen Pistorius, and Joe LoVetri, “On Super-Resolution With an Experimental Microwave Tomography System”, IEEE ANTENNAS AND WIRELESS PROPAGATION LETTERS, VOL. 9, pp.393-396, 2010). In this paper, the resolution was confirmed after estimating the dielectric constant accurately. A reviewer can't indicate you to enough literature on the reconstruction of the Born approximation. However, some papers show that small target with low-contrast can be accurately reconstructed by Born approximation.
https://www.mdpi.com/1424-8220/20/7/1905
https://opg.optica.org/oe/fulltext.cfm?uri=oe-14-19-8837&id=105666
https://engineering.purdue.edu/~malcolm/purdue/DiffractionTomographyThesis/thesis5.pdf
On the other hand, all reconstruction results in this paper are not satisfactory. The result is like a local minimum, and it seems that it is not suitable for estimating the resolution. Reconstruction was also performed with an object parameter (relative permittivity of 1.2 and 2) that is not suited to the Born approximation. If you discuss resolution with imprecise permittivity, you are saying that your method is not microwave tomography (Inverse electromagnetic scattering), it is radar.
